# Association between Adverse Reactions to the First and Second Doses of COVID-19 Vaccine

**DOI:** 10.3390/vaccines10081232

**Published:** 2022-07-31

**Authors:** Ken Goda, Tsuneaki Kenzaka, Shinsuke Yahata, Masanobu Okayama, Hogara Nishisaki

**Affiliations:** 1Hyogo Prefectural Tamba Medical Center, Department of Internal Medicine, 2002-7 Iso, Hikami-cho, Tamba 669-3495, Japan; kenkenpetneed@yahoo.co.jp (K.G.); honssk-d@sanynet.ne.jp (H.N.); 2Division of Community Medicine and Career Development, Kobe University Graduate School of Medicine, 2-1-5 Arata-cho, Hyogo-ku, Kobe 652-0032, Japan; 3Hyogo Prefectural Harima-Himeji General Medical Center, Department of General Internal Medicine, 3-264 Kamiya-cho, Himeji 670-8560, Japan; yahata-jci@umin.ac.jp; 4Division of Community Medicine and Medical Education, Kobe University Graduate School of Medicine, 2-1-5 Arata-cho, Hyogo-ku, Kobe 652-0032, Japan; okayamam@med.kobe-u.ac.jp

**Keywords:** COVID-19 vaccine, adverse reactions, first dose, second dose

## Abstract

This study investigated the frequency of adverse reactions to COVID-19 vaccines in Japan and the impact of first-dose adverse reactions on second-dose adverse reactions. Individuals who received an mRNA COVID-19 vaccine at our center in March or April 2021 were included. Data were collected using questionnaires. The main factors were age (<40, 40–59, and >60 years), sex, underlying disease, and first-dose adverse reaction. The primary outcomes were incidence of local and systemic adverse reactions (ARs) attributable to the vaccine. Logistic regression was used to calculate odds ratios (ORs) and 95% confidence intervals (CIs). Among 671 participants, 90% experienced local or systemic ARs. An AR to the first dose was associated with a significantly increased risk of an AR to the second dose (OR: 49.63, 95% CI: 21.96–112.16). ARs were less common among men than among women (OR: 0.36, 95% CI: 0.17–0.76). Local ARs were less common among those aged 60 years or older (OR: 0.35, 95% CI: 0.18–0.66), whereas systemic ARs were more common among those aged under 40 years. Information on ARs to the first dose is important for healthcare providers and recipients when making vaccination decisions.

## 1. Introduction

Vaccination against infectious diseases is a cost-effective method of disease prevention, and achieving herd immunity is important to protect those who cannot be vaccinated for various reasons [1]. While multiple doses of the vaccine against SARS-CoV-2 are available, adverse reactions to COVID-19 vaccines have become a matter of concern, and the public’s hesitancy toward the vaccine is a major issue [2].

Reports from Europe and the United States suggest that the SARS-CoV-2 vaccine is highly effective and has minor, but serious, side effects [3]. In Japan, the BNT162b2 messenger ribonucleic acid (mRNA) vaccine (Comirnaty, BNT162b2, BioNTech/Pfizer; Pfizer, New York, NY, USA) is being administered to healthcare workers. The mRNA vaccine is administered by intramuscular injection; upon receiving this injection, immune-competent cells, such as muscle and dendritic cells, produce an mRNA protein, and a portion of the protein is presented to the lymphocytes to elicit an immune response [4]. Aversion to side effects is widely recognized as a major reason for vaccine hesitancy, and transparent and independent evidence of safety are needed, especially for new vaccines [5].

In Japan, many reports of serious adverse reactions to COVID-19 vaccines have been published [6], but few studies have analyzed the frequency of adverse reactions [7]. Although adverse reactions are more frequently reported among women than among men and among younger people than among older people after immunization against COVID-19 [3,8], to our knowledge, no previous studies have investigated the impact of first-dose adverse reactions on second-dose adverse reactions. Clarifying the impact of first-dose adverse reactions on second-dose adverse reactions would help clinicians and recipients deal with adverse reactions and hesitancy toward receiving multiple doses. Hence, this study aimed to investigate the frequency of adverse reactions to the COVID-19 vaccine in Japan and the impact of first-dose adverse reactions on second-dose adverse reactions.

## 2. Materials and Methods

### 2.1. Study Design

This was a prospective cohort study. The study was approved by the Ethics Committee of the Hyogo Prefectural Tamba Medical Center (approval number: Tan-I number 1305). Participants provided written informed consent for the use of the questionnaires and for the publication of the study results.

### 2.2. Participants

In March and April 2021, in a mass vaccination program, a COVID-19 mRNA vaccine (Comirnaty, (BNT162b2, BioNTech/Pfizer; Pfizer, New York, NY, USA) was administered to the hospital staff at Tamba Medical Center. The vaccination was performed at their request and with their consent. Staff members who had received two doses of vaccine were included in the study.

### 2.3. Setting

Participants were given a self-administered questionnaire regarding adverse reactions to the vaccine, and a standardized questionnaire on general health status, which was used to obtain information on the individual’s basic attributes. The questionnaires were numbered and distributed in two batches. The vaccination questionnaire collected data about the situation on the day of vaccination and symptoms (adverse reactions) up to the 14th day post-vaccination. Only participants who had data on the first and second doses were included in the analyses.

### 2.4. Data Collected from the Questionnaire

Data on the following characteristics were collected: age, sex, pregnancy status (women only), underlying conditions, body temperature at the time of vaccination, physical condition at the time of vaccination, and food/drug allergy history. Data were collected on the following local adverse reactions at the injection site: redness, swelling, induration, pain, heat sensation, pruritus, and a feeling of heaviness. Data were collected on the following systemic adverse reactions: fever, chills, headache, fatigue, nasal discharge, cough, nausea, diarrhea, difficulty in moving the arms, and numbness.

### 2.5. Definition of Underlying Conditions

The presence of one or more of the following conditions was defined as an underlying condition: cancer, autoimmune diseases, diseases associated with steroid/immunosuppressant use, renal disease, diabetes, hepatic disease, and pulmonary disease.

### 2.6. Main Factors

The main factors were (1) no control groups with respect to age (<40 years, 40–59 years, and ≥60 years), sex, and underlying disease, and (2) no control group with respect to the first-dose adverse reaction.

### 2.7. Primary Outcomes

The primary outcomes were adverse reactions after the second doses of vaccine and included adverse reactions (“none” or “any one of local and systemic adverse reactions”); local adverse reactions (“none” or “any one of local adverse reactions”); and systemic adverse reactions (“none” or “any one of systemic adverse reactions”).

### 2.8. Statistical Analysis

Logistic regression analysis was used to calculate odds ratios (ORs) and 95% confidence intervals (CIs) for age, sex, underlying conditions, and the first-dose adverse reaction for each primary outcome measure. The analyses were performed using the following three models:Model 1—Unadjusted ORs.Model 2—Adjusted for sex, age, and underlying conditions.Model 3—Model 2 + adjusted for a first-dose adverse reaction.Stata MP version 16 (StataCorp, College Station, TX, USA) was used for all the analyses.

## 3. Results

The sample selection process is shown in the flowchart (Figure 1). Overall, 904 and 894 staff received a first and second dose of the vaccine, respectively (88.50% vaccination coverage); and 800 and 679 completed the first-dose and second-dose questionnaires, respectively (response rate: 75.95%). A total of 671 participants whose data on both vaccine doses were available were included in the analysis.

The basic attributes and the incidence of local and systemic adverse reactions in the 671 participants are shown in Table 1 (missing values were excluded from the analysis). The mean age ± standard deviation was 42.8 ± 15.1 years, and 74.8% of the participants were female. The majority of participants had no underlying conditions.

A total of 598 (89.1%)/564 (84.1%) participants had local reactions, 438 (65.3%)/537 (80.0%) had systemic reactions, and 628 (93.6%)/621 (92.6%) had local or systemic adverse reactions after the first and second doses, respectively.

Autoimmune conditions and diseases associated with steroid/immunosuppressant use included rheumatoid arthritis, hypo/hyperthyroidism, atopic dermatitis, and kidney transplant. After the first dose of vaccine, 22 (81.5%), 12 (44.4%), and 22 (81.5%) of the participants with autoimmune conditions or diseases associated with steroid/immunosuppressant use had local reactions, systemic reactions, and local or systemic adverse reactions, respectively. After the second dose of vaccine, 25 (92.6%), 14 (51.9%), and 25 (92.6%) of participants with autoimmune conditions or diseases associated with steroid/immunosuppressant use, had local reactions, systemic reactions, and local or systemic reactions, respectively.

Local adverse reactions after the first dose included pain at the vaccination site in 518 (79.0%), heaviness at the vaccination site in 283 (44.1%), and swelling at the vaccination site in 95 (14.8%) participants. Local adverse reactions after the second dose included pain at the vaccination site in 490 (73.9%), heaviness at the vaccination site in 329 (50.0%), and swelling at the vaccination site in 155 (23.8%) participants.

The most common systemic adverse reactions after the first dose were difficulty in moving the arms in 312 (48.1%), myalgia in 220 (33.8%), and fatigue in 84 (13.0%) participants. The most common systemic adverse reactions after the second dose were fatigue in 398 (60.8%), headache in 295 (45.1%), myalgia in 279 (42.8%), and difficulty in moving the arms in 269 (41.8%) participants. Fever occurred in only 23 participants (3.6%) after the first dose, and in 264 participants (40.7%) after the second dose.

If there was an adverse reaction after the first dose, the risk of any adverse reaction after the second dose was significantly higher (adjusted OR (aOR): 49.6, 95% CI: 21.96–112.16). Males were significantly less likely to experience an adverse reaction than females (aOR: 0.36, 95% CI: 0.17–0.76) (Table 2).

Among those aged 60 years and older, the aOR of developing local adverse reactions compared with those aged under 40 years was 0.35 (95% CI: 0.18–0.66) (Table 3). The reactions were significantly more likely to occur after the second dose of vaccine (aOR: 18.37 (95% CI: 8.68–38.86).

Participants with a systemic adverse reaction after the first dose of the vaccine were more than 10 times more likely to have a systemic adverse reaction after the second dose (aOR: 10.62, 95% CI: 5.13–21.99) (Table 4). Systemic adverse reactions were also less common in males than in females (aOR: 0.41, 95% CI: 0.27–0.64), and the risk of a systemic adverse reaction decreased with increasing age (Table 4).

## 4. Discussion

In this study, local or systemic adverse reactions following COVID-19 vaccination were reported by more than 90% of the participants. Participants with adverse reactions after the first dose were significantly more likely to experience adverse reactions after the second dose. Although it has been reported that adverse reactions are more common after the first dose than after the second one [3,8], to our knowledge, this is the first report to show that adverse reactions to the first dose of vaccine were associated with a substantially increased risk of adverse reactions to the second dose. In addition, the incidence of systemic adverse reactions was higher after the second dose than after the first dose. As in previous studies, local and systemic adverse reactions were significantly more likely to occur in women than in men, and in younger adults than in older adults [3,8].

In terms of the association between the first-dose and second-dose adverse reactions, the proportion of local reactions did not increase after the second dose, and none of the participants reported Grade 4 local reactions [3]. In general, most local reactions were mild-to-moderate and resolved within 1–2 days. However, the reported incidence of systemic adverse reactions was higher after the second dose than after the first dose, as in previous studies [3,8]. Izumo et al. [8] reported a higher incidence of adverse reactions, including a higher incidence of Grade 3 adverse reactions after the second dose than after the first dose; this study also examined the evolution of antibody titers after COVID-19 vaccination and found that only 35% of the participants had positive antibody titers after the first vaccination, and all of them showed positive titers after the second vaccination. These results suggest that two doses are necessary to obtain COVID-19 prophylaxis in the clinical setting [9]. Furthermore, after the second dose, the antibody titers were significantly negatively correlated with age. This means that older people have lower antibody titers than younger people, which may be associated with the lower incidence of adverse effects from vaccination in the older age groups. Therefore, there may be a correlation between antibody titers and adverse reactions [8].

In a study of healthy adults in Australia and the Philippines, the incidence of both local and systemic adverse reactions to the influenza vaccination was reported to be around 50% [10], but in a Japanese study of medical professionals, the incidence of local reactions was 73.9–81% [11]. However, in another Japanese study of medical professionals, the occurrence of local reactions was reported to be 73.9–81.7%, and that of systemic adverse reactions, 15.8–20.0% [11,12]. The COVID-19 vaccine showed a slightly higher frequency of local adverse reactions (89.1% for the first dose and 84.1% for the second dose). In a multinational clinical trial of the BNT162b2 vaccine, the incidence of serious adverse events was reported to be low and comparable between the vaccine and placebo groups [3]. However, when minor adverse reactions were included in the analysis, the overall incidence rates were 93.6% after the first dose and 92.5% after the second dose, and the incidence rates of systemic adverse reactions were slightly higher than those of local reactions at 65.3% and 80.0%, respectively. 

In our study, participants with autoimmune diseases or diseases associated with steroid/immunosuppressant use had a lower risk of systemic adverse reactions than other participants. Although there no participants in this study that had myasthenia gravis, vaccines against SARS-CoV-2 showed good short-term safety in myasthenia gravis patients, who may take advantage of the vaccination with regard to avoiding life-threatening complications, such as COVID-19 pneumonia [13]. After the first dose, 70.8% experienced adverse events, which consisted of local pain (76.5%), asthenia (29.4%), cephalalgia (17.6%), myalgia (5.9%), and with four patients (23.5%) experiencing two or more adverse reactions. No patients experienced fever after the first dose. There may be a correlation between antibody titers and adverse reactions [13].

The most common local adverse reactions caused by the influenza vaccine are injection site pain (75.6%), injection site swelling (18%), and injection site redness (10.4%) [11], similar to the common local adverse reactions caused by the COVID-19 vaccine. In contrast, the common systemic adverse reactions caused by the influenza vaccine include fatigue (8.7%), difficulty moving the arm (7.1%), and headaches (4.9%) [12], while the COVID-19 vaccine causes many other systemic adverse reactions; for example, in a study published in Germany, the most common systemic adverse reactions were headache/fatigue (48.1%), followed by myalgia (28.1%), fatigue (18.8%), arthralgia (14.3%), chills (13.9%), and fever (9.9%) [14]. The safety report of the Centers for Disease Control and Prevention on the BNT162b2 vaccine among volunteers in phase III trials included headache/fatigue (44.1%), myalgia (25.4%), chills (19.7%), arthralgia (15%), and fever (7.9%) as the most common systemic adverse reactions [15], which is similar to the findings of the current study.

In an international phase 2/3 study by Polack et al. [3], systemic adverse reactions were more common in younger (16–55 years) than in older (55 years and older) participants, and more common after the second dose of the vaccine than after the first dose. Fever (body temperature of 38 °C or higher) was reported to occur in 16% of the young and 11% of the older participants after the second vaccination and in 4% and 1%, respectively, after the first vaccination [3]; the median age in this study was 52 years, and 42% of the participants were aged 55 years or older. On the other hand, a Japanese study of healthcare workers reported an increase in the incidence of fever from 3.0% after the first dose to 44.9% after the second dose [8]. Our results (3.6% to 40.7%, respectively) were similar to a previous study [8]. This finding can be attributed to the fact that the definition of fever was 37.5 °C or higher, and the participants were relatively young. Other systemic symptoms, such as chills, headache, fatigue, and arthralgia, were also more common after the second dose.

In our study, local and systemic adverse reactions were less common among those aged 60 years or older, and systemic adverse reactions were more common among those younger than 40 years and 40–59 years. Other studies have also found that adverse reactions are more common in younger individuals than in older individuals [14,16]. Adverse reactions to vaccines are thought to be a byproduct of excessive production of type I interferon (IFN-I), which initiates an effective immune response against invading pathogens [17]. The production of IFN-I in women and young adults has been found to be greater than that in men and older adults [17,18], so this may explain the higher incidence of adverse reactions in women and younger adults than in men and older adults.

In the present study, some local and systemic adverse reactions occurred at a significantly lower rate in men. This is consistent with previous reports of more frequent adverse reactions in women [8,19,20,21]. The incidence of adverse reactions has been reported to be higher in women than in men after receiving mRNA-based and inactivated viral vaccines [19]. Women have consistently been shown to have an increased risk of adverse reactions after immunization with viral vaccines, such as influenza, measles-mumps-rubella combined vaccine, attenuated Japanese encephalitis vaccine, and attenuated dengue vaccine, and women tend to show vigorous immune responses [20].

Polyethylene glycol (PEG) has also been implicated as a cause of anaphylaxis in women [21]. Polyethylene glycol (PEG) is found in mRNA vaccines. PEG is also used in shampoos, toothpaste, and cosmetics. Women use cosmetics more frequently than men and have more exposure on a regular basis. Patients previously exposed to PEG may have high levels of antibodies to PEG and may be at risk for anaphylactic reactions to vaccines [21]. Although no anaphylaxis occurred in this study, these mechanisms may explain the increased incidence of adverse reactions in women and younger people.

Participants with autoimmune conditions and conditions associated with steroid/immunosuppressant use were less likely to develop systemic reactions compared with other participants. The second dose might represent a more vigorous response from crossing T cells or antibodies against antigens or anaphylaxis after a sensitization process. Therefore, healthcare providers should carefully collect information on reactions to the first dose so as to be prepared for adverse reactions after the second dose. This information may also help individuals to make informed decisions about vaccinations and help with the use of antihistamines, steroids, and other anti-allergy medications prior to vaccination.

### Limitations

We did not conduct regular observations to monitor the occurrence of adverse reactions, but collected data based on self-reports instead. As the participants were healthcare professionals, the reliability of the self-reports is likely to have been high, but the possibility of inconsistencies in reporting due to subjectivity cannot be ruled out. As the participants were medical professionals, there is also a possibility of selection bias. This may limit the generalizability of these results. The degree of adverse reactions was not quantified, and difference in the degree of adverse reactions after each dose was not assessed.

## 5. Conclusions

Vaccination with an mRNA-based COVID-19 vaccine caused local or systemic adverse reactions in approximately 90% of the study participants. Participants with an adverse reaction after the first dose were more than ten-fold more likely to experience an adverse reaction after the second dose. Adverse reactions were more common in women than in men. Local adverse reactions were less common among younger participants than older participants, and the risk of adverse reaction decreased significantly with increasing age. Healthcare providers should carefully collect information on reactions to the first dose, and should be prepared for adverse reactions after the second dose. These study results could also help individuals to make informed decisions about COVID-19 vaccination.

## Figures and Tables

**Figure 1 vaccines-10-01232-f001:**
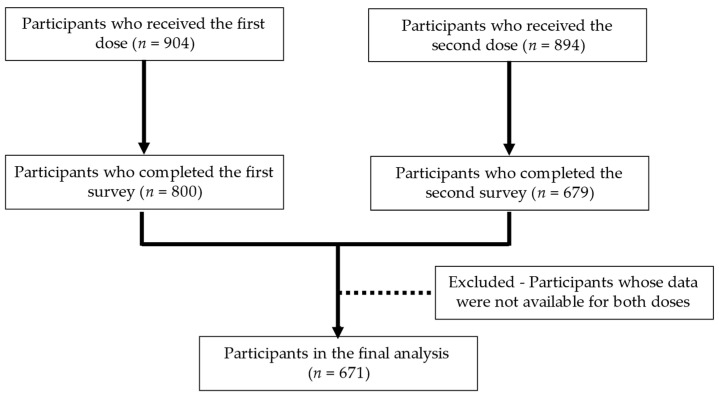
The flowchart of the sample selection process.

**Table 1 vaccines-10-01232-t001:** Participants’ background characteristics and symptoms in the 2 weeks after vaccination according to age.

	Overall				<40 Years				40–59 Years				60 Years or Older			
	First Vaccination		Second Vaccination		First Vaccination		Second Vaccination		First Vaccination		Second Vaccination		First Vaccination		Second Vaccination	
	*n* = 671				*n* = 274		*n* = 273		*n* = 295		*n* = 294		*n* = 100		*n* = 103	
	*n*	%	*n*	%	*n*	%	*n*	%	*n*	%	*n*	%	*n*	%	*n*	%
Sex																
Male	169	25.2	169	25.2	71	25.9	70	25.6	59	20.0	60	20.4	39	39.0	39	37.9
Female	502	74.8	502	74.8	203	74.1	203	74.4	236	80.0	234	79.6	61	61.0	64	62.1
Age (years: mean, SD)	42.8	15.1	42.8	15.1	27.0	6.2	27.0	6.1	49.9	5.8	49.8	5.7	64.7	4.2	64.6	4.2
Pregnant (female only)	0	0.0	5	1.1	0	0.0	2	1.0	0	0.0	1	0.5	0	0.0	2	3.2
Cancer/malignant tumor	3	0.4	3	0.4	0	0.0	0	0.0	1	0.3	1	0.3	2	2.0	2	1.9
Autoimmune conditions/diseases associated with steroid/immunosuppressant use	27	4.0	27	4.0	9	3.3	9	3.3	11	3.7	11	3.7	7	7.0	7	6.8
Renal disease	5	0.7	5	0.7	2	0.7	2	0.7	1	0.3	1	0.3	2	2.0	2	1.9
Diabetes	12	1.8	12	1.8	1	0.4	1	0.4	7	2.4	7	2.4	4	4.0	4	3.9
Hepatic disease	2	0.3	2	0.3	0	0.0	0	0.0	1	0.3	1	0.3	1	1.0	1	1.0
Pulmonary disease	15	2.2	15	2.2	8	2.9	8	2.9	6	2.0	6	2.0	1	1.0	1	1.0
Body temperature at the time of vaccination (°C: mean, SD)	36.4	0.3	36.4	0.3	36.5	0.3	36.4	0.3	36.4	0.3	36.4	0.3	36.3	0.3	36.2	0.4
Good physical condition at the time of vaccination	653	98.9	646	98.2	270	98.9	263	98.9	284	99.0	284	97.3	97	99.0	98	99.0
Food/drug allergies	68	10.2	64	9.6	28	10.3	25	9.3	29	9.9	28	9.5	11	11.2	11	10.7
Localized symptoms																
Redness at the injection site	57	8.8	71	10.9	31	11.5	32	12.3	20	7.0	33	11.6	6	6.4	6	5.9
Swelling at the injection site	95	14.8	155	23.8	48	18.0	79	30.2	32	11.4	56	19.6	15	16.3	19	18.8
Induration at the injection site	87	13.6	96	14.8	37	13.8	49	18.8	37	13.2	36	12.6	12	13.0	10	9.9
Pain at the injection site	518	79.0	490	73.9	222	82.2	210	78.1	235	81.3	220	75.3	59	62.1	59	58.4
Heat sensation at the injection site	87	13.6	139	21.4	48	18.1	75	28.3	32	11.5	52	18.2	7	7.4	12	12.0
Itching at the injection site	46	7.2	91	14.1	19	7.1	31	11.9	21	7.5	42	14.8	4	4.3	18	17.8
Heat sensation at the injection site	283	44.1	329	50.0	129	48.0	155	58.1	129	46.4	135	47.0	25	26.9	38	36.9
Swelling of the axilla on the side of inoculation	12	1.9	33	5.2	4	1.5	12	4.7	4	1.5	18	6.5	4	4.3	3	3.1
Other localized symptoms	26	4.5	34	6.0	7	2.9	10	4.4	15	5.9	19	7.5	4	4.8	5	5.6
Any localized symptoms	598	89.1	564	84.1	251	91.6	236	86.4	268	90.8	255	86.7	77	77.0	72	69.9
Systemic symptoms																
Fever	23	3.6	264	40.7	15	5.6	138	52.3	7	2.5	106	37.3	1	1.1	20	20.0
Chills	11	1.7	202	31.1	4	1.5	102	38.5	7	2.5	83	29.4	0	0.0	17	16.7
Headache	63	9.8	295	45.1	23	8.6	142	53.6	36	12.8	130	45.5	4	4.3	23	22.5
Fatigue	84	13.0	398	60.8	40	14.9	188	70.9	34	12.1	175	61.0	10	10.6	34	33.3
Nasal discharge	16	2.5	24	3.7	2	0.7	8	3.1	9	3.2	9	3.2	5	5.3	7	7.0
Cough	13	2.0	28	4.3	1	0.4	10	3.8	10	3.6	16	5.6	2	2.2	2	2.0
Nausea	12	1.9	62	9.5	3	1.1	41	15.6	9	3.2	21	7.3	0	0.0	0	0.0
Diarrhea	8	1.2	25	3.8	1	0.4	11	4.2	5	1.8	13	4.5	2	2.2	1	1.0
Myalgia	220	33.8	279	42.8	95	35.4	126	47.7	97	33.9	116	40.6	28	29.8	37	36.6
Arthralgia	27	4.2	203	31.1	8	3.0	93	35.1	15	5.3	96	33.6	4	4.3	14	13.9
Difficulty moving the arms	312	48.1	269	41.7	135	50.2	112	42.1	141	49.8	130	46.8	34	36.2	27	27.0
Numbness	23	3.6	35	5.4	8	3.0	13	4.9	13	4.6	20	7.0	2	2.2	2	2.0
Other systemic symptoms	28	4.5	50	8.2	7	2.7	10	4.0	18	6.6	33	12.4	3	3.4	7	7.6
Any systemic symptoms	438	65.3	537	80.0	188	68.6	232	85.0	197	66.8	236	80.3	51	51.0	68	66.0
Any localized or systemic symptoms	628	93.6	621	92.5	257	93.8	253	92.7	282	95.6	281	95.6	87	87.0	86	83.5

SD, standard deviation.

**Table 2 vaccines-10-01232-t002:** Factors associated with any adverse reaction (either local or systemic adverse reaction), after the second dose of COVID-19 mRNA vaccine.

	Model 1 OR (95% CI)	Model 2 aOR (95% CI)	Model 3 aOR (95% CI)
Sex			
Female	Reference	Reference	Reference
Male	0.23 (0.13–0.42)	0.25 (0.14–0.46)	0.36 (0.17–0.76)
Age			
<40 years	Reference	Reference	Reference
40–59 years	1.71 (0.83–3.51)	1.52 (0.73–3.16)	1.71 (0.71–4.14)
60 years or older	0.40 (0.20–0.80)	0.39 (0.18–0.83)	0.51 (0.20–1.32)
Underlying conditions			
No	Reference	Reference	Reference
Yes	1.06 (0.50–2.24)	1.71 (0.75–3.94)	1.08 (0.41–2.86)
Adverse reaction to first dose			
No	Reference	···	Reference
Yes	59.87 (27.66–129.58)	···	49.63 (21.96–112.16)

Model 1: unadjusted; Model 2: adjusted for sex, age, and underlying conditions; Model 3: Model 2 + adjusted for adverse reaction to first dose. Abbreviations: aOR, adjusted odds ratio; CI, confidence interval; OR, odds ratio.

**Table 3 vaccines-10-01232-t003:** Factors associated with local adverse reactions after the second dose of COVID-19 mRNA vaccine.

	Model 1 OR (95% CI)	Model 2 OR (95% CI)	Model 3 OR (95% CI)
Sex			
Female	Reference	Reference	Reference
Male	0.50 (0.32–0.77)	0.53 (0.34–0.83)	0.82 (0.44–1.22)
Age			
<40 years	Reference	Reference	Reference
40–59 years	1.03 (0.63–1.66)	0.94 (0.58–1.54)	0.92 (0.54–1.22)
60 years or older	0.36 (0.21–0.63)	0.32 (0.18–0.57)	0.35 (0.18–0.66)
Underlying conditions			
No	Reference	Reference	Reference
Yes	1.27 (0.73–2.22)	1.90 (1.03–3.50)	1.63 (0.85–3.10)
Adverse reaction to first dose			
No	Reference	···	Reference
Yes	21.45 (10.38–44.34)	···	18.37 (8.68–38.86)

Model 1: unadjusted; Model 2: adjusted for sex, age, and underlying conditions; Model 3: Model 2 + adjusted for adverse reaction to first dose. Abbreviations: aOR, adjusted odds ratio; CI, confidence interval; OR, odds ratio.

**Table 4 vaccines-10-01232-t004:** Factors associated with systemic adverse reactions after the second dose of COVID-19 mRNA vaccine.

	Model 1 OR (95% CI)	Model 2 aOR (95% CI)	Model 3 aOR (95% CI)
Sex			
Female	Reference	Reference	Reference
Male	0.34 (0.23–0.51)	0.35 (0.23–0.53)	0.41 (0.27–0.64)
Age			
<40 years	Reference	Reference	Reference
40–59 years	0.72 (0.46–1.12)	0.64 (0.41–1.01)	0.61 (0.38–0.98)
60 years or older	0.34 (0.20–0.58)	0.33 (0.19–0.58)	0.37 (0.20–0.67)
Underlying conditions			
No	Reference	Reference	Reference
Yes	1.01 (0.62–1.64)	1.48 (0.87–2.53)	1.29 (0.74–2.23)
Adverse reaction to first dose			
No	Reference	···	Reference
Yes	13.17 (6.55–26.49)	···	10.62 (5.13–21.99)

Model 1: unadjusted; Model 2: adjusted for sex, age, and underlying conditions; Model 3: Model 2 + adjusted for adverse reaction to first dose. Abbreviations: aOR, adjusted odds ratio; CI, confidence interval; OR, odds ratio.

## Data Availability

The data sets used and/or analyzed during the present study are available from the first author on reasonable request.

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
