# Peer review of "Association between Adverse Reactions to the First and Second Doses of COVID-19 Vaccine"

_vaccines, 2022, doi:10.3390/vaccines10081232_

Round 1
Reviewer 1 Report
Here, Goda et al reported a study to investigate the impact of potential adverse reactions of the first does mRNA COVID-19 vaccination on that of the second dose vaccination in Japan, with 671 participants included. Intriguingly, their analysis revealed that adverse reactions are less common among men than among women. The earlier reports always suggested the opposite directional impact. Lastly, they found that the first-dose adverse reactions correlated with the second-dose adverse reactions. This correlation is significant.
This is a manuscript targeting on a important vaccine issue faced for the field, since adverse reactions associated with vaccination is one of the major concerns for people with the vaccine hesitancy. This study provided scientific information to address this concern. This study also provided information to provide the base for healthcare providers to design vaccination administration plans to avoid the potential adverse effects associated with following boost. This is particularly valuable since there is ongoing policy for repeated vaccinations.
Major criticism:
The current content needs to be presented in a more logic way, which will be much easier to follow.
Author Response
Thank you for the valuable comments. We have added the following sentence in the introduction: “Although adverse reactions are more frequently reported among women than among men and among younger people than among older people after immunization against COVID-19 [3,8],” (Pages 1-2, Lines 46-48). In addition, the manuscript has been edited by an English editor for clarity, consistency, and logical flow, and we have made the terminology consistent throughout.
We have replaced Table 1 and removed “No” and “Unknown.” We have also reformatted the tables, added more precise table titles, and shown the reference groups in Tables 2, 3, and 4. In addition, we used various terms to refer to study participants, including participants, respondents, and patients. In the revised manuscript, the term “participants” has been used consistently throughout.

Reviewer 2 Report
This study investigates the frequency of adverse reactions to the COVID-19 vaccine in Japan and the impact of the first-dose adverse reactions on the second-dose adverse reactions. The paper sounds interesting, quite organized and comprehensive. Vaccination against COVID-19 has raised many concerns in public opinion. I think that it is a very relevant topic that must be addressed. The design of the study is good. I only have some suggestions:
- This study examines data from 671 participants examining comorbidities in general populations and different ages; however, some special populations should be discussed. What about patients affected by autoimmune disease? Consider the role of vaccination in special populations and autoimmune neuromuscular disease, such as myasthenia gravis. Read and cite recent papers facing this relevant topic (https://doi.org/10.3390/neurolint14020033).
-Adverse reactions were less common among men than among women; worsen outcome from vaccination and systemic symptoms were reported in the female gender and after the first dose. This is in line with recent reports that showed higher incidence of AEs in the female gender and after the first dose (https://doi.org/10.3390/neurolint14020033). The same applies for the young, who experience higher AEs rates, if compared with older people. The authors discuss the role of Polyethylene glycol as a cause of anaphylaxis in women, but I wonder if there might be a role for autoimmune comorbidities in distinguishing people who are more prone to develop systemic AEs. On this perspective, the second dose might represent a more vigorous response from crossing T-cells or antibodies against antigens or anaphylaxis after a sensitization process. Is there a relevant risk by repetition of vaccines in these populations? The authors should also discuss this point because the role of PEG may be quite marginal.
-the design of the study is in line with the aims. The results have been discussed in full.
-there are no relevant grammar mistakes.
Author Response
Response: Thank you for the valuable comments. Special populations were not the focus of this study; thus, we do not consider it appropriate to discuss them in detail. We have added some discussion on patients with autoimmune diseases or diseases associated with steroid/immunosuppressant use (Page 6, Lines 123-131, and Page 8, Lines 211-220). In addition, we have added reference 13 (Page 8, Line 216, 220 and Page 11, Lines 347-349).
Response: Thank you for the valuable comments. We have revised the text as follows: “Polyethylene glycol (PEG) has also been implicated as a cause of anaphylaxis in women [21]. Polyethylene glycol (PEG) is found in mRNA vaccines. PEG is also used in shampoos, toothpaste, and cosmetics. Women use cosmetics more frequently than men and have more exposure on a regular basis. Patients previously exposed to PEG may have high levels of antibodies to PEG and may be at risk for anaphylactic reactions to vaccines [21]. Although no anaphylaxis occurred in this study, these mechanisms may explain the increased incidence of adverse reactions in women and younger people.
Participants with autoimmune conditions and conditions associated with steroid/immunosuppressant use were less likely to develop systemic reactions compared with other participants. The second dose might represent a more vigorous response from crossing T cells or antibodies against antigens or anaphylaxis after a sensitization process. Therefore, health care providers should carefully collect information on reactions to the first dose so as to be prepared for adverse reactions after the second dose. This information may also help individuals make informed decisions about vaccinations.” (Page 9-10, Lines 265-278).

Round 2
Reviewer 1 Report
The updated manuscript has addressed my main concern.